# Evaluation of Multiple Diagnostic Indicators in Comparison to the Intestinal Biopsy as the Golden Standard in Diagnosing Celiac Disease in Children

**DOI:** 10.3390/medsci4040020

**Published:** 2016-11-25

**Authors:** Elisabet Hollén, Malin Farnebäck, Tony Forslund, Karl-Eric Magnusson, Tommy Sundqvist, Karin Fälth-Magnusson

**Affiliations:** 1Division of Microbiology and Molecular Medicine, Department of Clinical and Experimental Medicine, Faculty of Medicine and Health Sciences, Linköping University, SE-581 85 Linköping, Sweden; tony.forslund@liu.se (T.F.); karl-eric.magnusson@liu.se (K.-E.M.); tommy.sundqvist@liu.se (T.S.); 2Dynamic Code AB, SE-582 56 Linköping, Sweden; Malin@dynamiccode.se; 3Division of Paediatrics, Department of Clinical and Experimental Medicine, Linköping University, SE-581 85 Linköping, Sweden; karin.falth-magnusson@liu.se; 4Department of Paediatrics and Department of Clinical and Experimental Medicine, Linköping University, SE-581 85 Linköping, Sweden

**Keywords:** celiac disease, children, diagnosis, small bowel biopsy, nitric oxide, antibodies, genetic analyses of HLA type and SNPs

## Abstract

Celiac disease (CD) is a chronic small intestinal enteropathy triggered by gluten in genetically predisposed individuals. The susceptibility is strongly associated with certain human leukocyte antigen (HLA)-genes, but efforts are being made in trying to find non-HLA genes that are predictive for the disease. The criteria for diagnosing CD were previously based primarily on histologic evaluation of small intestinal biopsies, but nowadays are often based only on blood tests and symptoms. In this context, we elucidated the accuracy of three diagnostic indicators for CD, alone or in combination. Genetic analyses of HLA-type and nine single nucleotide polymorphisms (SNPs) known to be associated with CD were performed in 177 children previously investigated for the suspicion of CD. CD was confirmed in 109 children, while 68 were considered non-celiacs. The antibodies and urinary nitrite/nitrate concentrations of all of them were measured. The combinations of all the variables used in the study would classify 93% of the study population in the correct diagnostic group. The single best predictors were antibodies (i.e., anti-endomysium immunoglobulin A (IgA) (EMA) and transglutaminase IgA (TGA)), followed by HLA-type and nitric oxide (NO)-metabolites. The nine SNPs used did not contribute to the right diagnoses. Although our control group consisted of children with mostly gastrointestinal symptoms, the presented methodology predicted a correct classification in more than 90% of the cases.

## 1. Introduction

Celiac disease (CD) is a chronic small intestinal enteropathy which is immune-mediated and triggered by ingestion of dietary gluten in genetically predisposed individuals [1]. It is one of the most common chronic diseases in childhood, with a prevalence ranging between 1% [2] and 3% [3]. CD often presents itself during early childhood, lately with a trend towards a delayed onset at six or seven years of age [4], and is characterised by a variety of clinical symptoms, specific serum antibody responses, and a spectrum of damages and immunological aberrations to the small intestinal mucosa. The only known treatment of CD is life-long adherence to a strict gluten-free diet (GFD).

Moreover, CD is a polygenetic disorder, and the susceptibility is strongly associated with certain human leukocyte antigen (HLA) alleles. The HLA heterodimers DQ2 and DQ8 have been shown to represent the highest genetic risk for CD, with almost 100% of CD patients carrying one or both of these alleles. However, these genes are represented in approximately 40% of the population, making the positive predictive value poor [5]. Several genome-wide association studies (GWAS) have been performed in order to identify non-HLA genes contributing to the genetic susceptibility of CD [6,7], and new risk regions have been proposed—most of which contain genes that control the adaptive immune response. Some of the regions are also associated with other autoimmune disorders, such as type 1 diabetes and Crohn’s disease [8,9].

Nitric oxide (NO) is a free radical produced from l-arginine by the nitric oxide synthases (NOS). In untreated CD, there is increased excretion of NO metabolites (i.e., of nitrite and nitrate) into the urine [10]. After institution of a GFD, the NO metabolites decrease accordingly [10], but increase again after gluten challenge [11]. This is presumably caused by increased activation and expression of iNOS (inducible NO synthase) in the mucosa of CD patients [12], and the elevated NO levels in the urine likely indicate an on-going inflammation in the small intestine.

The criteria for the diagnosis of CD in children have been changed over time. The former 1990 guidelines of the European Society for Paediatric Gastroenterology, Hepatology, and Nutrition (ESPGHAN) [13] were based on the histologic evaluation of the small intestinal biopsy as the golden standard, as accompanied by CD-specific antibody tests and symptoms. The antibody tests were first based on anti-gliadin immunoglobulin A (IgA) (AGA), later on anti-endomysium IgA (EMA), and after an autoantigen in CD was discovered [14], on anti-tissue transglutaminase IgA (TGA). The new guidelines launched by ESPGHAN in 2012 [15] say that in some cases the diagnosis of CD may be done without the need of a biopsy; i.e., when there are clear symptoms, positivity for HLA DQ2 and/or DQ8, and TGA levels >10 times the cut-off level.

In this context, we aimed at elucidating the accuracy of three different non-invasive index tests (i.e., analyses of genetic markers, serum antibodies, and urinary nitrite/nitrate) in diagnosing CD. We found that the combination of genetics, serum antibodies against tissue transglutaminase, and urinary NO metabolites provided the best congruence with the reference standard (i.e., the pathologist evaluation of the intestinal biopsy).

## 2. Materials and Methods

### 2.1. Study Population

Letters containing information on the study, together with materials to perform sampling of buccal epithelial cells (i.e., buccal swabs) were sent out to 196 children who had been investigated for the suspicion of CD at three paediatric clinics in southeast Sweden between 1996 and 2007, and hence who were diagnosed according to the old criteria [13], where biopsies were taken under anaesthesia with a capsule and graduated [12] according to established criteria by the pathologist. The children were randomly chosen from our local files on celiac children in southeast Sweden. The inclusion criteria were: less than 18 years of age, known antibody titres to EMA and/or TGA, a urinary sample for the measurement of urinary nitric oxide levels collected at the time of the investigation, and a biopsy-proven diagnosis as being either CD or non-CD.

### 2.2. Antibody Analyses

Antibody titres were determined as part of the standard diagnostic procedure at the Department of Clinical Immunology at the Linköping University Hospital for AGA, EMA, TGA, or a combination of two or three of the antibodies. No IgA-deficient individuals were included.

The EMA titres were given as the biggest dilution factor for a positive signal (e.g., 1/640), which is why we used the denominators for the calculations—in the example, 640. AGA and TGA were given as units/mL (U/mL).

### 2.3. DNA Diagnostics

The DNA diagnostic method was focused on the analysis of nine single nucleotide polymorphisms (SNPs) (Table 1) previously reported to have an association with CD [7,16,17,18,19]. In addition, HLA-DQ2.5 and/or HLA-DQ8 associations were determined by analysing nucleotides 553 in HLA-DQA1 and 185, 242, 266, and 304 in HLA-DQB1.

DNA was extracted from buccal swabs using the ZR-96 genomic DNA Kit (Zymo Research, Irvine, CA, USA) according to the manufacturer’s protocol. DNA fragments surrounding the SNPs, and part of HLA-DQA1 and HLA-DQB1 genes were amplified with Ex-Taq (Takara Bio Inc., Shiga, Japan). SNP and HLA-DQ analyses were performed using SNaPshot™ analysis (Life Technologies, Stockholm, Sweden). The SNaPshot products were quantified on a 310 Genetic Analyzer (Thermo Fisher Scientific, Waltham, MA, USA).

For the calculations, we coded the SNP genotypes as “0” for the negative homozygotes, “1” for the heterozygotes having one of the risk alleles, and “2” for the homozygotes with both the risk alleles. The HLA genotypes were designated accordingly into three groups: “0” if they were negative for both DQ2.5 (DQA1*05 and DQB1*02) and DQ8 (DQB1*0302); “1” if they were heterozygous positive for either DQA1*05 or DQB1*02; and “2” if they were positive for DQ2.5 and/or DQ8.

### 2.4. Urinary NO Metabolites

In the urinary samples, the sum of nitrite and nitrate (NO_2_/NO_3_) was taken as an indirect indicator of the NO production [20]. In short, in a phosphate-buffered saline (PBS)-diluted sample, nitrate was converted to nitrite using nitrate reductase from *Aspergillus*, where 50 µL of the diluted urine was first mixed with 10 µL of nicotinamide adenine dinucleotide phosphate (NADPH) (1 µM) and then with 40 µL of nitrate reductase (80 U/L, Roche, Basel, Switzerland), glucose-6-phosphate (500 µM), and glucose-6-phosphate dehydrogenase (160 U/L). The reaction mixture was incubated at room temperature for 45 min, and then used for the Griess assay of nitrite by adding 100 µL sulfanilamide (1% in 5% phosphoric acid) and 100 µL naphtylethylenediamine (0.1%). The resultant colour was read at 540 nm with a spectrophotometer (Vmax, Molecular Devices, Sunnyvale, CA, USA).

### 2.5. Ethics and Statistics

All participants gave their informed consent, and the study was approved by the Human Research Ethics Committee at the Faculty of Medicine and Health Sciences, Linköping University, Sweden. The project was approved by the regional ethics committee on 28 January 2009, and was given the number M102-7 T10-09. Differences between the study groups were analysed with the two-tailed Student’s *t*-test. The contribution of the different variables to the overall percentage of correct classification was performed with logistic regression and discriminant analyses, in all cases using SPSS (SPSS Statistics for Windows, version 20.0; IBM Corp, Armonk, NY, USA), and *p*-values smaller than 0.05 were considered significant.

## 3. Results

In total, 177 children met the inclusion criteria of the study and returned the buccal swabs. Among them, 109 had previously been diagnosed as having CD (mean age 6.3 years, range 0.7–17.9 years, 48 male, 61 female) and 68 as non-celiacs (mean age 4.8 years, range 0.7–16.4 years, 39 male, 29 female). The latter individuals were all investigated on the suspicion of CD, and the reasons for referral were most commonly diarrhoea, abdominal pain, and/or other gastrointestinal symptoms. Heredity and failure to thrive were also common causes for an investigation. However, a presumption of CD was discarded in these patients after a thorough investigation, including serological and histological evaluation of small intestinal biopsies, as described in the next section.

### 3.1. Small Intestinal Biopsies

The diagnoses were defined through a combination of repeated small intestinal biopsies, using capsule sampling and serum antibodies, which have varied over the time span of this study. Three different grading systems were employed for the classification of the mucosal damage: Alexander grade (CD *n* = 64, no CD *n* = 67), Marsh classification (CD *n* = 29, no CD *n* = 3), and the “kvalitet standardisering” (KVAST) evaluation by the Swedish Society of Pathology (CD *n* = 52, no CD *n* = 5) [21,22]. For the calculations in the present study, we—together with an independent, experienced paediatrician—reclassified the material as follows: biopsy score 1 if there was a normal mucosa (Alexander I, Marsh 0, and KVAST normal); biopsy score 2 for patients with mild enteropathy (Alexander II, Marshs I and II, and KVAST borderline); and finally, biopsy score 3 when there was a severe enteropathy (Alexanders III and IV, Marsh IIIa, IIIb, and IIIc, and KVAST partial or subtotal/total villous atrophy). In the CD group, two children had biopsy score 2, and all the others had biopsy score 3 at the time of the diagnosis. In the non-celiac group, only one had score 2, while all the others had biopsy score 1.

### 3.2. Antibody Analyses

Due to the time span when the participants in this study were diagnosed, the serology tests in clinical use varied. AGA was tested in 151 children, EMA in 121, and TGA in 138 children. A combination of two or three of the antibody tests was available in 153 children. AGA was only used in combination with either EMA or TGA. No IgA-deficient individuals were seen.

#### 3.2.1. AGA

The mean concentrations of AGA were 74.8 U/mL and 47.3 U/mL for the CD (*n* = 88) and the non-CD (*n* = 63) groups, respectively (*p* < 0.003). Using AGA alone, with a cut off-value at 30 U/mL, 45 CD children would have been falsely identified as negative, giving a sensitivity of 49%. The specificity of AGA was 57%, with 27 non-CD children regarded as falsely positive.

#### 3.2.2. EMA

The mean value of the EMA-titres (dilution factors) was 912.0 for the CD group (*n* = 67) and 9.1 for the non-CD group (*n* = 53) (*p* < 0.001). The cut off-value used was 10 (e.g., 1/10), giving a sensitivity of 93% and a specificity of 89% using EMA alone, resulting in five CD children with falsely negative and six non-celiacs with falsely positive diagnoses.

#### 3.2.3. TGA

The mean concentration of TGA was 80.8 U/mL for the CD group (*n* = 89) and 5.8 U/mL for the non-CD group (*n* = 49) (*p* < 0.001). Using a cut off-value at 7 U/mL yielded a sensitivity of 91% for TGA, with eight CD children falsely negative. Four non-CD children appeared as falsely positive using this TGA test alone, giving a specificity of 92%. See Table 2 for a comparison of the three antibody tests used.

### 3.3. Urinary NO Metabolites

Urinary samples were collected from all children in the study. The mean ± standard error of the mean (SEM) urinary nitrite/nitrate concentrations were 2483 ± 232 μM for the CD group and 940 ± 89 μM for the non-CD group (*p* < 0.001) (Figure 1). The cut off-value was set at 1500 μM, according to a previous study [10]. Accordingly, in the present study, 50 CD children displayed concentrations smaller than 1500 μM, giving a sensitivity of the method of 54%. The specificity was 90%, with only seven non-CD children with nitrite/nitrate levels higher than 1500 μM (Table 2).

### 3.4. DNA Diagnostics

#### 3.4.1. HLA Variants

Among the CD children 85% were DQ2 positive only, 3% had only DQ8, and 6% were both DQ2 and DQ8 positive. The remaining 6% were either DQ8 positive together with DQA1*05, one of the alleles for DQ2 (2%), positive only for one of the alleles DQA1/DQB1 (2%), or negative for both DQ2 and DQ8 (2%).

For the non-CD children, 32% were DQ2 positive only, 10% DQ8, and 3% positive for both DQ2 and DQ8, while only one patient had DQ8+DQA1*05. One of the alleles for DQ2 (DQA1/DQB1) was found in 18% of the non-celiac individuals, and 35% of them were negative for both DQ2 and DQ8 (Figure 2).

#### 3.4.2. Non-HLA Variants

The distribution of risk alleles for the nine SNPs used was uneven in the two study groups, but no clear difference between celiacs and non-celiacs could be seen. Characteristics, distribution, and odds ratio (OR) for each SNP are presented in Table 1.

### 3.5. Contribution of the Variables to a Correct Diagnosis

In order to assess the overall contribution of each of the variables tested in the study to a correct diagnosis, we used two different approaches: logistic regression and discriminant analysis.

#### 3.5.1. Logistic Regression

Nitrite/nitrate concentrations together with antibodies and HLA-type provided the best combination to predict CD without biopsy, with 93.2% of the study population given the right diagnosis. In the CD group, two children were placed in the non-CD group, while ten non-celiacs were incorrectly placed in the CD group. The nine SNPs used did not contribute to the diagnosis, since their inclusion in the regression test did not improve the number of children correctly diagnosed. Moreover, the nine SNPs alone gave only 64.8% overall percentage of correct diagnoses; i.e., with 20 in the CD group and 42 in the control group being mismatched, respectively. By contrast, antibodies being positive or negative were the single best predictors of CD, with 92.7% being correctly diagnosed, and the addition of HLA type did not improve that number (Table 3).

#### 3.5.2. Discriminant Analysis

Assessing the correlation between the discriminating biopsy score variable and standardized canonical discriminant function showed that antibodies contributed most to a correct classification with a correlation of 0.92. HLA-type and nitric oxide metabolites yielded weaker correlations of 0.40 and 0.24, respectively. The correlations of the SNPs were very weak, showing both positive and negative values, the absolute values ranging between 0.046 and 0.001 (Figure 3).

By contrast, 92.6% were correctly classified, resulting in eight false positive and five false negative results, which equalled that for antibodies alone. So, adding the other parameters did not improve the overall percentage of correctly classified individuals (Table 4).

Using TGA alone (138 children), the predicted accuracy was 80%, and when combined with NO and HLA, it increased to 85.5%. For AGA or EMA (120 children) alone, the corresponding values were 53% and 75%, respectively (i.e., lower than for TGA—data not shown).

## 4. Discussion

The golden standard in the diagnosis of celiac disease (CD) has been based on three small intestinal biopsies: the first taken at presentation and referral to normal diet, the second after a period of GFD, and the third upon gluten challenge. However, CD may now be assessed on the basis of symptoms and serology only, saving time, resources, and the patients’ inconvenience. It is important that new non-invasive methods are both sensitive and specific enough to reflect the damage and inflammation of the intestinal mucosa that is characteristic to CD. The new ESPGHAN guidelines [15] state that if the patient displays symptoms indicative of CD and the TGA levels are more than 10 times higher than the normal values, then the diagnosis may be confirmed without biopsies, but by further testing for EMA antibodies and HLA-typing. We therefore aimed at retrospectively assessing—in a child population that was previously diagnosed based on the classical criteria—the diagnostic accuracy of the serology; i.e., of AGA, EMA, and TGA antibodies, HLA-typing, and of two non-invasive methods (non-HLA genetics from buccal swabs and urinary NO metabolites).

We found that the combination of the variables used in this study would classify 93% of our study population in the right diagnostic groups, using either logistic regression or discriminant analyses. Among these, the antibody tests of EMA—and especially TGA—contributed most to the classification values. TGA had the best specificity, and EMA had the best sensitivity, both leaving approximately 10% of the study population incorrectly classified. Interestingly, when doubling the cut-off levels, we got no false positives, but an increase of false negatives. A ×10 higher cut-off resulted in even more false negative CD patients, confirmed by another study where almost 30% of the CD patients were regarded as negative [23]. When testing four different TGA-assays, Vermeersch et al. [24] found differences in disease likelihood between the assays at ×10 above the cut-off, indicating that statistical probability tests might be more accurate than multiples of cut-off values when deciding whether or not to diagnose without histologic evaluations.

Adding HLA-DQ genotypes and concentrations of NO metabolites in the urine to the calculations did not improve the overall percentage of correctly classified individuals. The small contribution of HLA-type was likely due to the high incidence of DQ2 and/or DQ8 in the population (around 40%) [5]. Urinary nitric oxide metabolites had a high specificity of 90% for CD, but the sensitivity was only 54%, reducing the usefulness of the test as a sole diagnostic tool. We have previously shown that the excretion of NO metabolites in the urine follows the disease activity [10,11,25], and hence the level of inflammation in the intestine, making it valuable as an instrument for follow up of the GFD on an individual basis, but not for the diagnosis as such.

To assess the usefulness of non-HLA variants, we employed SNPs from regions previously proposed in the literature to be associated with CD [7,16,17,18,19]. However, adding them to the discriminant analyses made no difference in sensitivity or specificity, and the contribution of an individual SNP to the correct classification was poor, indicating a lack of diagnostic accuracy. Again, we focused on only nine SNPs covering five different loci, as compared to a recent study by Romanos et al. [26] where 57 SNPs were used to calculate the CD risk prediction. They showed that by adding the non-HLA to the HLA risk variants, the sensitivity increased only from 35% to 63%, with a decreased positive predictive value. Furthermore, among diabetic children developing CD, Sharma et al. [27] tested 48 SNPs proposed to be associated with CD, and they found that only one SNP (rs13015714/IL18RAP) provided a significant difference.

Taken together with our results, non-HLA variants known hitherto, together with HLA typing, may strengthen the suspicion of CD, but they are in our opinion of little diagnostic value when standing alone. We would, however, bring forward the new method for sampling of genetic material through buccal swabbing, which is very practical. It is simple, user-friendly, and can be used for other applications.

## 5. Conclusions

In conclusion, our approach to combine serology, genetics, and urinary NO metabolites showed a good agreement with the classical, invasive histologic evaluation of the small intestinal mucosa in CD. It should be stressed that our reference group comprised children referred to the clinic due to gastrointestinal symptoms and/or growth disturbances leading to the suspicion of CD; i.e., it was not a healthy, normal population, but represented a clinically relevant situation. Still, our methodological approach enabled a correct classification in more than 90% of the cases. Regardless, we still suggest that the small intestinal biopsy should not be fully eliminated, since it can yield important additional information on the structure and immunological status of the mucosa, both at presentation and following gluten-free diet treatment.

## Figures and Tables

**Figure 1 medsci-04-00020-f001:**
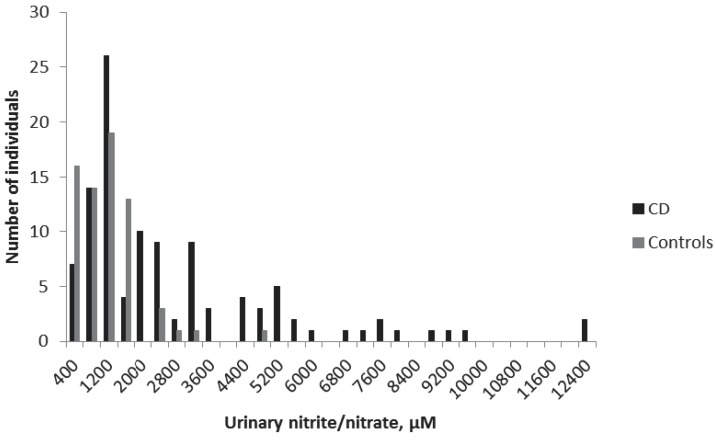
Concentrations of urinary nitric oxide metabolites in children with celiac disease (CD, *n* = 109, black) and control children (No CD, *n* = 68, grey).

**Figure 2 medsci-04-00020-f002:**
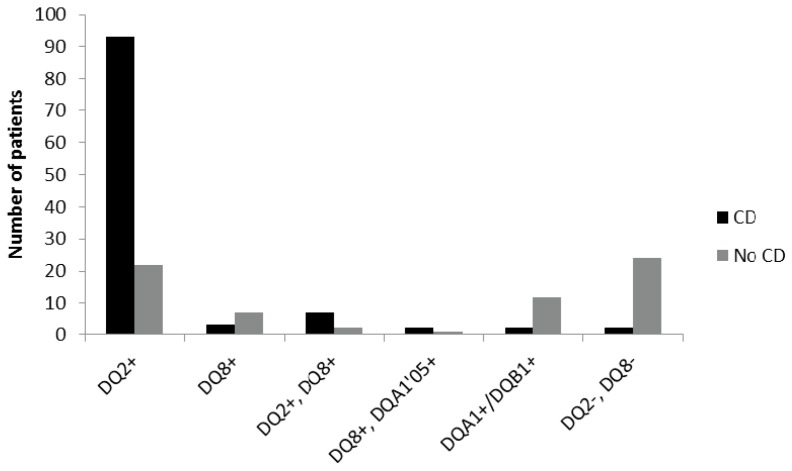
Number of children with celiac disease (CD, *n* = 109, black) and control children (No CD, *n* = 68, grey) with the different human leukocyte antigen (HLA)-types.

**Figure 3 medsci-04-00020-f003:**
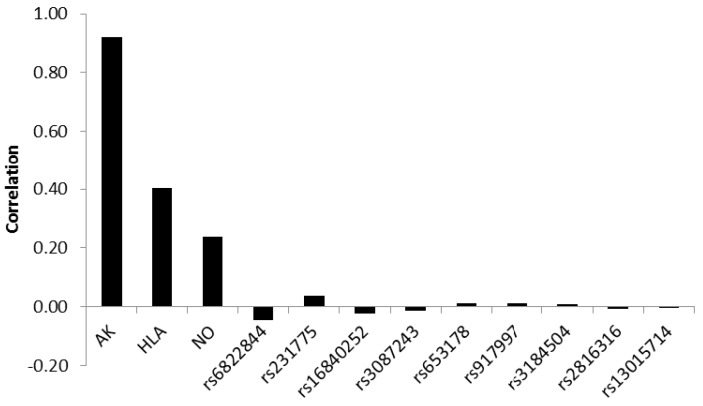
Correlations between the discriminating biopsy score variable and canonical discriminant functions in children with (*n* = 109) and without (*n* = 68) celiac disease. Ab: antibodies.

**Table 1 medsci-04-00020-t001:** Single nucleotide polymorphisms (SNPs) used for the DNA diagnostic method and the distributions of risk alleles in the study groups CD (Celiac disease) and No CD.

SNP	Locus	Location	Risk Allele	CD			No CD			OR ^d^
				0 ^a^	1 ^b^	2 ^c^	0	1	2	
rs2816316	*RGS1*	1q31	G	77	27	5	46	20	2	0.869
rs13015714	*IL18RAP*	2q11-2q12	G	55	41	13	30	34	4	0.775
rs917997			A	54	45	10	33	31	4	0.960
rs231775	*CTLA4*	2q33	G	30	56	23	17	44	7	0.878
rs3087243			A	41	47	21	22	34	12	0.793
rs16840252 *			T	75	29	4	45	20	3	0.861
rs6822844	*IL2/IL21*	4q27	T	84	23	2	47	20	1	0.666
rs3184504	*SH2B3, ATXN2*	12q24	T	33	47	29	19	34	16	0.893
rs653178			G	37	45	27	18	40	10	0.701

^a^ 0, negative for the risk allele for that SNP; ^b^ 1, heterozygous for the risk allele; ^c^ 2, homozygous for the risk allele; ^d^ OR, odds ratio; * Result for this SNP was missing for one of the CD patients.

**Table 2 medsci-04-00020-t002:** Specificity and sensitivity of the different tests used.

Test	False Positive, *n* (%)	False Negative, *n* (%)	Specificity, %	Sensitivity, %
AGA	27 (43)	45 (51)	57	49
EMA	6 (11)	5 (7)	89	93
TGA	4 (8)	8 (9)	92	91
NO	7 (10)	50 (46)	90	54

AGA, IgA anti-gliadin antibody; EMA, IgA anti-endomysium antibody; TGA, IgA anti-tissue transglutaminase antibody; NO, urinary nitric oxide metabolites.

**Table 3 medsci-04-00020-t003:** Predicted values in the logistic regression for the different variables and the overall predictive accuracy of correct classification into the CD (*n* = 109) or non-CD (*n* = 68) groups.

Variable(s)	False Positive, *n* (%)	False Negative, *n* (%)	Predictive Accuracy, %
HLA	32 (47.0)	4 (3.7)	79.7
Antibodies (ab)	8 (11.8)	5 (4.6)	92.7
NO	7 (10.3)	51 (46.8)	67.2
SNP	42 (61.8)	20 (18.3)	64.8
HLA + ab	8 (11.8)	5 (4.6)	92.7
HLA + ab + NO	10 (14.7)	2 (1.8)	93.2
HLA + ab + NO + SNP	7 (10.3)	6 (5.5)	92.6

**Table 4 medsci-04-00020-t004:** Results of the discriminant analyses showing sensitivity, specificity, and the predictive accuracy, when using different numbers of variables.

Variable(s)	False Positive, *n* (%)	False Negative, *n* (%)	Specificity, %	Sensitivity, %	Predictive Accuracy, %
HLA	32 (47.0)	4 (3.7)	59.2	96.3	79.7
Antibodies (ab)	8 (11.8)	5 (4.6)	88.2	95.4	92.7
NO	6 (8.8)	55 (50.5)	91.2	49.5	65.5
SNP	32 (47.0)	42 (38.5)	52.9	61.1	58.0
HLA + ab	8 (11.8)	5 (4.6)	88.2	95.4	92.7
HLA + ab + NO	8 (11.8)	5 (4.6)	88.2	95.4	92.7
HLA + ab + NO + SNP	8 (11.8)	5 (4.6)	88.2	95.4	92.6

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
