# Peer review of "Evaluation of Multiple Diagnostic Indicators in Comparison to the Intestinal Biopsy as the Golden Standard in Diagnosing Celiac Disease in Children"

_medsci, 2016, doi:10.3390/medsci4040020_

Round 1
Reviewer 1 Report
This is an interesting study evaluating multiple diagnostic indicators in comparison to the intestinal histology in diagnosing celiac disease in children. The study is well planned and performed, but a few inaccuracies limit its scientific impact.
1. Over the last two decades, it has been widely reported that in the presence of selective IgA deficiency, a condition strictly related to celiac disease, the antibodies AGA, EMA and TGA are not detectable if only the IgA isotype is tested. If the authors are not able to provide data on AGA, EMA and/or TGA of IgG isotype, which could improve the performance obtained in their work, this should at least be reported as a limitation of the study.
2. Among the keywords, genetic analysis of HLA-type and nine SNPs is missing.
3. In materials and methods, it is not reported where and eventually how the upper endoscopy with biopsy sampling and duodenal histology has been performed. Such information should be included in the study.
Author Response
Concerning Q1: No IgA- deficient individuals were diagnosed, which has been clarified in the revised manuscript.
Q2: The missing key word have been added.
Q3: Information on the technique for obtaining the biopsies has now been included. The conclusions have now been revised.
Reviewer 2 Report
I would like to thanks all authors for this paper titled :Evaluation of multiple diagnostic indicators in comparison to the intestinal 2 biopsy as the golden standard in diagnosing celiac disease in children. The authors in this paper aimed to elucidate the accuracy of three different non-invasive index tests, i.e. analyses of genetic markers, serum antibodies and urinary nitrite/nitrate, in diagnosing CD.They found that the combination of genetics, serum antibodies against tissue transglutaminase, and urinary NO metabolites provided the best congruence with the reference standard, i.e. the pathologists evaluations of the intestinal biopsy.
In my personal opinion these findings are really interesting for the future diagnostic guidelines of Celiac Disease particurarly in the children. But of course we need more numbers among all groups of people considered in this study.
Author Response
Unfortunately, we cannot increase the number of individuals to be included in the study, as there is a practical limit on how long a study can be extended.
Reviewer 3 Report
The paper is well written, the authors investigate about the combination of new markers for CD diagnosis, notwithstanding the aim and conclusions of this work are not clear enough.
Moreover, as shown in table 4, the suggested diagnostic approach does not provide any distinctive contribution to reach a correct diagnosis. the combination of different variables considered (HLA+Ab+NO) does not improve the specificity and sensitivity of antibody detection alone.
I suggest the authors to revaluate the work and submit these data as a brief comunication.
Author Response
We have tried to clarify the methodological issues raised and altered the conclusions.
A resubmission as a short communication was not requested by the Editor.
Although the results displayed in Table 4 do not provide “positive” support for using the new markers tested, we think that these “negative” findings might be even more valuable for focusing the diagnosis of celiac disease. In the Results section, we have also added information on the predictive values for AGA, EMA and TGA alone, and the latter in combination with the analyses of HLA and NO. Those figures have not been included in Table 4 for the sake of clarity, but recognized also in the Discussion on line 269 in the revised manuscript.
Round 2
Reviewer 3 Report
I agree with the Editor decision and with the other referees' suggestions.